# Mapping Attenuation Determinants in Enterovirus-D68

**DOI:** 10.3390/v12080867

**Published:** 2020-08-08

**Authors:** Ming Te Yeh, Sara Capponi, Adam Catching, Simone Bianco, Raul Andino

**Affiliations:** 1Department of Microbiology and Immunology, University of California, San Francisco, San Francisco, CA 94158, USA; mingte.yeh@ucsf.edu (M.T.Y.); benjamin.catching@ucsf.edu (A.C.); 2Industrial and Applied Genomics, AI and Cognitive Software, IBM Almaden Research Center, San Jose, CA 95120, USA; sara.capponi@ibm.com (S.C.); sbianco@us.ibm.com (S.B.); 3Center for Cellular Construction, University of California, San Francisco, CA 94158, USA; 4Graduate Group in Biophysics, University of California, San Francisco, San Francisco, CA 94158, USA

**Keywords:** enterovirus, enterovirus-D68, virulence determinant, mouse model, infectious clones, VP3, paralysis

## Abstract

Enterovirus (EV)-D68 has been associated with epidemics in the United Sates in 2014, 2016 and 2018. This study aims to identify potential viral virulence determinants. We found that neonatal type I interferon receptor knockout mice are susceptible to EV-D68 infection via intraperitoneal inoculation and were able to recapitulate the paralysis process observed in human disease. Among the EV-D68 strains tested, strain US/MO-14-18949 caused no observable disease in this mouse model, whereas the other strains caused paralysis and death. Sequence analysis revealed several conserved genetic changes among these virus strains: nucleotide positions 107 and 648 in the 5′-untranslated region (UTR); amino acid position 88 in VP3; 1, 148, 282 and 283 in VP1; 22 in 2A; 47 in 3A. A series of chimeric and point-mutated infectious clones were constructed to identify viral elements responsible for the distinct virulence. A single amino acid change from isoleucine to valine at position 88 in VP3 attenuated neurovirulence by reducing virus replication in the brain and spinal cord of infected mice.

## 1. Introduction

Enterovirus D68 (EV-D68), previously known as rhinovirus 87 [1], is a single-stranded, plus-sense RNA virus belonging to the genus *Enterovirus* of the *Picornaviridae* family. The viral genome encodes a polyprotein with a single open reading frame with untranslated regions at both ends. Secondary structures form an internal ribosome entry site (IRES) in the 5′- untranslated region (UTR) and mediate virus translation [2]. The polypeptide contains a P1 region that encodes the structural proteins (VP1, VP2, VP3 and VP4) and P2 and P3 regions that encode non-structural proteins (2A, 2B, 2C, 3A, 3B, 3C and 3D) that are important to virus replication. Although genetically closer to enteroviruses, EV-D68 possesses key characteristics of rhinovirus: a lower optimal growth temperature at 33 °C and acid sensitivity which limits its survival through the alimentary tract. EV-D68 is regarded as a respiratory pathogen that causes symptoms including sneezing, cough, runny nose and, in some cases, wheezing and difficulty breathing.

EV-D68 was first isolated from children with pneumonia and bronchiolitis in California in 1962 [3] but has remained silent for the past few decades. Starting with sporadic clusters of acute respiratory infections around the world in the 2000s [4,5,6,7], EV-D68 caused an outbreak with at least 1395 confirmed cases in the US in 2014 [8] and was associated with acute flaccid paralysis [9]. Infections were also reported in Canada, Argentina, Norway, Netherlands, France, Germany and Taiwan [10,11,12,13,14,15,16,17]. EV-D68, thus emerged as a neuro-invasive viral pathogen as it caused epidemics in 2016 and 2018 in the US [18,19] and worldwide [14,15,20,21,22,23,24].

Several animal models have been utilized to study the pathogenesis and pathogenicity of EV-D68. A ferret nasal infection model recapitulated lower respiratory tract pathogenesis and revealed increased levels of inflammatory cytokines and chemokines that may contribute to the cause of pulmonary edema [25]. Neonatal immune-competent mice are susceptible to EV-D68 infection with observed interstitial pneumonia or paralysis [26,27]. Adult immune-deficient AG129 mice also reproduce EV-D68 induced paralytic myelitis [28]. Notably, the distinct virulence of EV-D68 strains was observed in these studies. By adapting the EV-D68 strain US/MO/14/18949 in adult AG129 mice with 30 serial passages, several genetic changes throughout the viral genome resulted in increased virulence [28], but the precise virulence determinant is unknown in these mouse-adapted EV-D68 strains. Thus, the molecular determinant of EV-D68 virulence is unknown.

In this study, we tested the susceptibility of Tg21/IFNR-ko (Tg21 with type I interferon receptor knockout) mice to EV-D 68. These mice were used in our previous studies for poliovirus Sabin strains and Zikavirus [29,30,31]. We found that this mouse model is susceptible to EV-D68 and also recapitulates the paralysis observed in severe human infections, and different EV-D68 isolates show distinct virulence in this model. We then applied reverse genetics to generate chimeric and mutant EV-D68 viruses to identify the genetic element responsible for the distinct virulence.

## 2. Materials and Methods

### 2.1. Cells and Viruses

Rhabdomyosarcoma (RD, TCC CCL-136, Manassas, VA, USA) cells were maintained in Dulbecco’s modified Eagle’s medium (DMEM, Cell Culture Facility, University of California, San Francisco, CA, USA) supplemented with 10% fetal bovine serum (FBS, Sigma, St. Louis, MO, USA) and 1X penicillin/streptomycin (Gibco, Gaithersburg, MD, USA) at 37 °C with 5% CO_2_. SK-N-SH (ATCC HTB-11) and Neuro-2A (ATCC CCL-131) cells were maintained in Eagle’s minimum essential medium (EMEM, UCSF CCF) supplemented with 10% FBS and 1X penicillin/streptomycin under the same condition as above.

EV-D68 strains used in this study are listed in Table 1. Strains US/MO/14-18947, US/MO-14-18949 and US/IL/14-18952 were obtained through BEI Resources (NIAID, NIH, Bethesda, MD, USA) and used for construction of infectious cDNA clones. EV-D68 strain CA/14-4231 (4231) was provided by Dr. Charles Chiu (UCSF, San Francisco, CA, USA). Viruses used in experiments in Figure 1 and Figure 2 were prepared by amplifying the obtained viruses once in RD cells, and infectious cDNA clone-derived viruses were used in all other experiments (see Section 2.5). Since we constructed infectious cDNA clones for original, chimeric and point-mutated EV-D68 viruses (see Section 2.4), BEI Resources catalog numbers (49129, 49130, and 49131) for EV-D68 strains were used for easier differentiation.

Infectious cDNA clones of EV-D68 strains 49129, 49130 and 49131 have been banked in BEI Resources (catalog numbers: NR-52009, NR-52010 and NR-52011).

### 2.2. Ethical Statement

All animal experiments were conducted in accordance with the guidelines of the Laboratory Animal Center of National Institutes of Health. The Institutional Animal Care and Use Committee of University of California, San Francisco, approved all animal protocols on January 28, 2019 (approved protocol number AN178420-01A).

### 2.3. Animals

C57BL/6-derived Tg21 mice with type I interferon receptor knockout (Tg21/IFNR-ko) [32] were obtained from Dr. Satoshi Koike (Tokyo Metropolitan Institute of Medical Science, Tokyo, Japan). Mice were bred in-house in sterilized cages and maintained in a 12/12 light cycle with standard chow diet in the AAALAC-certified animal facility at UCSF. Specific pathogen-free (SPF), 5-10-day-old mice were used in this study.

### 2.4. Construction of EV-D68 Infectious cDNA Clones, Wild-Type, Chimeric and Point-Mutated

EV-D68 viruses were obtained from BEI Resources (Table 1) and used to extract viral RNA for cDNA synthesis. Viral RNA was extracted with ZR Viral RNA Extraction Kit (ZYMO Research, Irvine, CA, USA), following the manufacturer’s protocol. To construct full-length infectious cDNA clones of wild-type strain 49129, RT-PCR was performed using the Superscript III One-Step RT-PCR System with Platinum Taq DNA Polymerase (Invitrogen, Carlsbad, CA, USA) with primers 1FEcoRI and RT30SalI to amplify viral genome. PCR product was gel-purified, digested with *EcoRI* and *SalI* (New England Biolabs, Ipswich, MA, USA) and ligated to pre-digested pBR322 vector (New England Biolabs, USA). For wild-type strains 49130 and 49131, viral genomic fragments were amplified as described above, with primer sets 1F-InFus/R3390 and 3376F/RT30InFus for strain 49130 and 1F-InFus/R3808 and 3795F/RT30InFus for strain 49131, and cloned into pUC19 (New England Biolabs, USA) by using In-Fusion HD Cloning Kit (Takara Bio USA, Mountain View, CA, USA). Full-length sequences of the resulting infectious cDNA clones were confirmed with Sanger sequencing and used for our sequence analysis. Sequences of primers are provided in Appendix A.

Infectious cDNA clones of chimeric EV-D68 were constructed by replacing 5′-UTR or P1 gene of strains NR-49131 and 49129 with corresponding sequences of strain 49130. Briefly, viral gene fragments were PCR amplified from each wild-type infectious cDNA clone, gel-purified and cloned into pUC19 by using In-Fusion HD Cloning Kit (Takara Bio, USA).

Nucleotide and amino acid variations between virulent and non-virulent strains were introduced into the infectious cDNA clone of strain 49131 by PCR-mutagenesis. Plasmid DNA of strain NR-49131 was PCR amplified with primers carrying desired nucleotide or amino acid change. PCR product was digested with DpnI (New England Biolabs, USA) and then transformed into SURE 2 competent cells (Agilent Technologies, Santa Clara, CA, USA). These sequence variations were introduced individually into infectious cDNA clones of strain 49131, except VP1-K282R and VP1-G283E, which were introduced together.

### 2.5. Generation of Viruses from Infectious cDNA Clones

Infectious viral RNA was transcribed from *SalI*-linearized plasmid DNA with MEGAscript T7 Transcription Kit (Invitrogen), following the manufacturer’s instruction, and examined on a standard agarose gel for RNA integrity. Viral RNA was electroporated into RD cells with a BTX electroporator (BTX, Holliston, MA, USA) using the following settings: 300 V, 1000 µF, 24 Ω in a 0.4 cm electro-cuvette (BTX). Electroporated cells were incubated 24 h at 33 °C and then frozen at −80 °C and thawed at room temperature 3 times to release the virus. Cell lysate was cleared at 1500× g for 10 min under 4 °C and supernatant was stored as passage 0 (P0) virus stock. RD cells were infected with P0 virus at m.o.i. (multiplicity of infection) of 0.1 at 33 °C for 24 h and harvested as described above for the P1 virus used in this study. Virus titers were determined by a standard TCID_50_ assay [33] and calculated with the Spearman and Karber method, as previously described [34]. Full-length viral genomic sequence was confirmed with Sanger sequencing.

### 2.6. Infection of Mice

Tg21/IFNR-ko mice were intraperitoneally (i.p.) or intracranially (i.c.) inoculated with either wild-type, chimeric, mutant viruses or viral media as a mock control group. I.p. inoculation was performed by injecting 100 µL of inoculum delivering 5 × 10^2^–1 × 10^5^ TCID_50_ of virus per mouse (5–9 mice per virus dosage). I.c. inoculation was performed to deliver 1 × 10^3^–1 × 10^4^ TCID_50_ of virus in a 10-µL inoculum per mouse (7–11 mice per virus dosage). Mice of 5, 8 and 10 days old were used to test susceptibility to EV-D68 and 7 days old for identification of virulence determinant. Neurovirulence test for VP3-I88V mutant virus was performed with 5-day-old mice. Infected mice were monitored daily for signs of disease onset including ruffled hair, hunched back, reduced mobility and paralysis. Mice were euthanized upon appearance of paralysis on both posterior limbs, the humane endpoint.

### 2.7. Tissue Distribution of Virus in Mice

Seven-day-old Tg21/IFNR-ko mice were i.p. inoculated with 100 µL of inoculum carrying 1 × 10^5^ TCID_50_ of wild-type or mutant EV-D68 virus. For i.c. inoculation, 10 µL of inoculum was injected to deliver 1 × 10^4^ TCID_50_ of wild-type or mutant EV-D68 virus. At days 1, 3 and 5 post-inoculation, 3 mice from each virus group were euthanized, perfused with 1X PBS (UCSF CCF), and selected tissues were collected aseptically, weighted and stored at −80 °C. Tissue samples were homogenized in viral medium, disrupted by 3 freeze-thaw cycles and cleared at 1500× *g* for 10 min at 4 °C. Muscle, brain and spinal cord were harvested from i.p. and i.c. infected mice, while blood, spleen, heart, kidney, small intestine and liver were harvested only from i.p. inoculated mice. Virus titer in cleared supernatant was determined by a standard TCID_50_ assay [33] and calculated with the Spearman and Karber method, as previously described [34].

### 2.8. Virus Replication Analysis

RD, SK-N-SH and Neuro-2A cells were infected with wild-type or mutant EV-D68 at m.o.i. (multiplicity of infection) of 0.01 for 1 h at 33 °C and then covered with growth medium. Cells were harvested at 0, 24, 48 and 72 h post-infection by freezing at −80 °C. Cells were frozen and thawed 3 times and cleared at 1500× *g* for 10 min under 4 °C. Virus titers in supernatant were determined by a standard TCID_50_ assay [33] and calculated with the Spearman and Karber method, as previously described [34]. Experiment was performed in triplicate and results shown as mean with S.D. (standard deviation).

### 2.9. Statistical Analysis

Data preparation and statistical analysis were performed with Prism 8 (Graphpad). Log-rank test was used to compare EV-D68 virulence in mouse model. Student’s *t*-test was performed to compare virus titers at collected time points. All multiple test correction was performed using the Holm–Sidak Method [35]. Statistical significance was defined as a *p* value less than 0.05.

### 2.10. Molecular Dynamics Simulations

To carry out the simulations of the wild-type EV-D68 protomer (VP1, VP2, VP3 and VP4) and its I88V mutant, we employed the atomic coordinates extracted from the human enterovirus D68 crystal structure (PDB ID 6CV2) [36]. A number of residues from the crystal structure [36] in the beta sheets’ connecting link were unresolved and we modeled these into the structure using CHARMM-GUI software [37]. We provided as input to CHARMM-GUI software the pdb-format structure file downloaded from the Protein Data Bank website (https://www.rcsb.org/); we patched the N-terminus of each capsid protein protomer with an acetylated (ACE) group and the C-terminus with a standard C-terminus patching group and modeled the missing residues by checking a box in CHARMM-GUI (CHARMM-GUI uses GalxyFill). To generate the mutated structure, we repeated this protocol and selected the “Mutation” box within CHARMM-GUI. We used CHARMM-GUI [37] for modeling the missing residues of the EV-D68 protomer crystal structure and for generating the I88V mutant. We used the same procedure to simulate the two systems and described it in the following. We solvated the system and added ions to maintain charge neutrality. To minimize the system, we used the conjugate gradient algorithm for 8000 steps and gradually heated the simulated cell from 25 to 300 K. To equilibrate the system, we ran 1 short NVT (constant number of particles N, volume V and temperature T) and 4 short NPT (constant number of particles N, pressure P and temperature T) simulations of 1 nanosecond (ns) in which we applied harmonic restraints to the protein backbone, water and ions. Such restraints were released gently during the consecutive runs. After the last run of the equilibration procedure, we carried out a simulation run of 1 ns in the NPT ensemble. All simulations were performed using NAMD 2.13 [38], with the CHARMM36m force field for the protein and ions [39] and the TIP3P model for water [40]. We used a Langevin dynamics scheme to keep the temperature constant at 300 K and anisotropic coupling in conjunction with the Nosé–Hoover–Langevin piston algorithm to keep the pressure constant at 1 atm [41,42]. Periodic boundary conditions were applied in three dimensions. We employed the smooth particle mesh Ewald (PME) summation method to calculate the electrostatic interactions [43,44] and the short-range real-space interactions were cut off at 10 Å using a switching function between 8 and 10 Å. The equations of motion were integrated with a time step of 4 femtosecond (fs) for the long-range electrostatic forces, 2 fs for the short-range non-bonded forces and 1 fs for the bonded forces by means of a reversible, multiple time-step algorithm [45]. The SHAKE algorithm [46] was used. Coordinates were saved every 20 picoseconds (ps). The simulations were visualized using VMD software [47].

## 3. Results

### 3.1. Tg21/IFNR-ko Mice Are Susceptibile to EV-D68 Infection

To test the susceptibility of Tg21/IFNR-ko mice to EV-D68, 5-, 8- and 10-day-old mice were inoculated with various dosages (5 × 10^2^–5 × 10^4^ TCID_50_ per mouse) of EV-D68 strain 49129 (US/MO/14-18947) via the i.p. route. Survival of 5-day-old infected mice was 0% for all three tested virus doses, and a delayed onset of death was observed at the lowest dose (5 × 10^2^ TCID_50_) (Figure 1A). While 5 × 10^3^ and 5 × 10^2^ TCID_50_ of virus led to 33% and 100% survival for 8-day-old mice (Figure 1B), only the highest dose, 5 × 10^4^ TCID_50_, of virus caused disease and a 0% survival in 10-day-old mice (Figure 1C). The survival of infected mice correlated with mouse age and virus dosage. In addition, infected mice showed paralysis on hindlimbs (Figure 1D), recapitulating the neurovirulence of EV-D68. We also tested the susceptibility of 7-day-old, immunocompetent Tg21 mice to EV-D68 and found no observable disease with an i.p. injection of up to 1 × 10^5^ TCID_50_ of virus (data not shown). Thus, neonatal Tg21/IFNR-ko mice are susceptible to EV-D68 and we set out to use this mouse model to study EV-D68 pathogenicity.

### 3.2. EV-D68 Strains Showed Differential Virulence in Tg21/IFNR-ko Mice

To compare the virulence of our collected EV-D68 strains (listed in Table 1), 7-day-old Tg21/IFNR-ko mice were infected i.p. with 1 × 10^4^ TCID_50_ of strains 49129, 49130, 49131 or CA/14-4231. Strains 49129, 49131 and CA/14-4231 caused death between days 5 and 9 post-infection and resulted in a survival percentage between 26% and 37.5% (Figure 2A). On the other hand, strain 49130 caused no observable disease in this mouse model. We also used replication analysis to clarify whether the non-pathogenic phenotype of strain 49130 was due to impaired replication ability and found that it replicated as efficiently as the pathogenic strains (Figure 2B). Although some significant titer differences were found between strains 49129 and 4231 at 24, 48 and 72 h; between 49129 and 49131 at 24 h; between 49129 and 49130 at 24 h, and between 49131 and 4231 at 48 h, all these EV-D68 strains replicated relatively efficiently in RD cells, with ~3 log increase in virus titers after 24 h of infection. Thus, the non-pathogenic phenotype of strain 49130 in mice was not due to loss of replication ability. We then focused on mapping the viral genetic elements responsible for the distinct virulence.

### 3.3. Nucleotide Changes in the 5′-UTR Are Not Involved in EV-D68 Virulence

To identify the potential virulence determinant, sequences of NR-49129, NR-49130, NR-49131 and CA/14-4231 were aligned to reveal conserved genetic changes that we defined as nucleotides in 5′-UTR and amino acids in the coding region that are present only in the non-pathogenic strain but not in pathogenic ones (Appendix A). In the viral 5′-UTR, two positions were revealed as the conserved nucleotide difference: cytosine:uridine (pathogenic:non-pathogenic strain) at position 107 and adenine/guanine:cytosine at position 648. In addition, seven conserved amino acid differences in four viral proteins were revealed. These conserved genetic changes are listed in Table 2.

To locate viral genetic elements causing the distinct virulence, we generated several chimeric EV-D68 viruses by swapping viral genes between the virulent strains 49131 and 49129 and the non-virulent strain 49130 and also viruses carrying various single mutations to better characterize their effect on virulence attenuation. We started with the 5′-untranslated region (UTR) and generated a chimeric virus, 30-5UTR-49131, in which the 5′-UTR of strain 49131 was replaced with the sequence from strain 49130 and also two viruses carrying single mutation, C107U or G648C, in the 5′-UTR with 49131 as backbone (Figure 3A). Replication analysis confirmed that all these engineered viruses showed ~3 log increase in virus titer, suggesting efficient replication in RD cells (Figure 3B). The amount of injected virus was increased to 1 × 10^5^ TCID_50_ to better identify attenuation determinant. By i.p. infecting 7-day-old Tg21/IFNR-ko mice with 1 × 10^5^ TCID_50_ of virus, no mouse infected with 49131, 30-5UTR-49131, 49131-C107U or 49131-G648C survived after day 7 post-injection. We constructed infectious clones carrying the same swapped 5′-UTR and single mutation with 49129 as a template and generated viruses to repeat this experiment. We obtained similar results (data not shown), suggesting that the 5′-UTR of EV-D68 was not involved in the attenuation phenotype of 49130 (Figure 3C).

### 3.4. Amino Acid Change VP3-I88V Determined EV-D68 Virulence in Mice

Next, we examined the effect of amino acid variations in the coding regions on EV-D68 virulence by generating engineered EV-D68 viruses carrying swapped P1 or VP1 gene or the mutations described above (Figure 4A). Mutations in the non-structural region (NSR), 2A-T22A and 3A-H47R, were introduced to the pathogenic strain 49131. Cell-culture replication of the two these viruses was similar to the parental strains 49131 and 49130 (Figure 4B). However, while 49131-2A-T22A was as virulent as 49131, around 40% of mice survived infection with 49131-3A-H47R (Figure 4C).

Since multiple amino acid variations within the structural region of the EV-D68 genome were identified, we generated P1 or VP1 swapped viruses to test their effects on virulence (Figure 5A). Replication analysis with RD cells and a virulence test with Tg21/IFNR-ko mice were also performed, as described above. Both the VP1 and P1 swapped viruses replicated efficiently in RD cells, with a 2–3 log increase in virus titers after 24 h of infection, although virus titers were 1–2 log lower than for strains 49131 and 49130 at several time points (Figure 5B). In addition, virus 30VP1-49131 had virulence comparable to 49131 in mice and caused 0% survival for the infected mice, and 30P1-49131 caused no disease, like 49130 (Figure 5C). The attenuation by the P1 gene of 49130 was confirmed by swapping the P1 gene of 49129: the resulting 30P1-49129 virus caused 100% survival and no observable disease in infected mice (data not shown). This result suggested that the mutation I88V in VP3 is responsible for the attenuation phenotype of 49130 since VP1 swap had little effect on virulence.

To test effects of the mutations in structural gene on virulence, we introduced these mutations (VP3-I88V, VP1-L1P, VP1-V148A and VP1-K282R/G283E) into 49131 for replication and virulence tests. Similar to the VP1 or P1 swapped viruses, the mutant viruses 49131-VP3-I88V and 49131-VP1-V148A replicated in RD cells with comparable dynamics to the parental strain 49131. A 2 log increase in virus titer was observed at 24 h post infection for the 49131-VP1-L1P and 49131-VP1-K282R/G283E mutants. At later time points, 49131-VP1-L1P replicated to titers comparable to that of 49131 at later time points, but no further increase in virus titer was observed for 49131-K282E/G283E, suggesting its significant impact on virus replication (Figure 5D). Results from the mouse virulence test showed that mutations VP1-V148A and VP1-K282R/G283E did not affect virulence as they caused death in 100% of infected mice within 7 days. While mutation VP1-L1P allowed a 50% survival of infected mice, VP3-I88V did not cause disease, suggesting its role as major virulence determinant for EV-D68 in this mouse model (Figure 5E).

To further confirm the potent attenuation effect on virulence by the mutation VP3-I88V, a neurovirulence test was performed by injecting 5-day-old mice with 49131 or 49131-VP3-I88V via the i.c. route. Survival rate was 0% for mice infected with 1 × 10^4^ TCID_50_ of 49131 and 18.2% for the lower dose of 1 × 10^3^ TCID_50_ of virus with delayed death. Consistent with i.p. inoculation, 49131-VP3-I88V caused no disease at both tested doses (Figure 6).

These results suggest that the amino acid change of VP3-I88V, VP1-L1P and 3A-H47R reduced EV-D68 virulence in mice, and, among them, VP3-I88V showed the strongest effect. Thus, we further characterized the VP3-I88V mutant.

### 3.5. VP3-I88V Reduced Virus Replication in Mice

To determine why VP3-I88V attenuated EV-D68 virulence, the tissue distribution of the injected virus was examined by i.p. inoculating 7-day-old mice with 1 × 10^5^ TCID_50_ of 49131 or 49131-VP3-I88V virus. Various tissues were collected at days 1, 3 and 5 post-infection for a TCID_50_ assay to determine virus titers. Virus 49131 replicated efficiently, with an increase of 3.3 logs in muscle, 5.1 in spinal cord and ~2 logs in brain during the 5 days, but no significant virus replication was observed in other collected tissues (Figure 7A). On the other hand, virus titers of 49131-VP3-I88V increased 1.5 log in muscle, 0.7 log in spinal cord and was detected at a very low titer without notable increase in the brain (Figure 7A).

To further clarify whether VP3-I88V mutation affected viral entry from peripherals to the central nervous system (CNS) or viral ability to replicate in the CNS, viruses 49131 or 49131-VP3-I88V were i.c. injected directly into the brains of 5-day-old mice. Brain, spinal cord and muscle were collected at days 1, 3 and 5 post-infection. Virus 49131 replicated extensively, with an increase in virus titer of 2.6 logs in brain and 3.9 logs in spinal cord during the 5 days, but it was detected at very low titer without a significant increase in muscle. Virus 49131-VP3-I88V was detected at very low titers in these tissues (Figure 7B). Results from tissue distribution analysis suggest that the VP3-I88V mutation reduced the replication ability of EV-D68 in mouse tissues and, more importantly, in the CNS.

### 3.6. EV-D68 49131 and VP3-I88V Mutant Virus Showed Comparable Replication Ability in Cell Lines

Since 49131 and the 49131-VP3-I88V mutant showed distinct replication ability in mice, we examined their replication in cultured cells. RD (human rhabdosarcoma cell), SK-N-SH (human neuroblastoma cell) and Neuro-2A (mouse neuroblastoma cell) were infected at m.o.i. of 0.01 and harvested at 0, 24, 48 and 72 h post-infection for titration. The viruses replicated with comparable kinetics in RD, SK-N-SH and Neuro-2A cells, suggesting that the VP3-I88V mutation had no effect on the replication ability of EV-D68 in these tested cells (Figure 8).

### 3.7. Comparison of the Simulated Capsid Structures of EV-D68 VP3 and the VP3-I88V Mutant

To gain insights into the reduced virulence of EV-D68 caused by the I88V mutation in VP3, we turned to full-atom molecular dynamics (MD) simulations, which allow examination at atomistic resolution of the effects of this amino acid change on the structure. VP1, VP2 and VP3 capsid proteins (protomers) associate to form a pentamer (Figure 9A). The protomers share a wedge-shaped, eight-stranded, antiparallel, beta-barrel topology where the connecting loops are structurally different, depending on the specific protomer to which they belong [48,49,50]. VP3-I88 is located in the CD loop of VP3, between the C beta strand and the αA helix (Figure 9A), in proximity to the interface between VP2 and VP3 of two different promoters and of the VP1 C-terminus. The I88 points inward, toward the αA helix and the cavity enclosed by the eight antiparallel beta strands (Figure 9B). Such geometry favors interactions with R104 in the αA helix and with F178 in the GH loop, which likely makes contact with VP2 of the adjacent protomer (Figure 9A, inset). Interestingly, R104 interacts with D232 and D91, which are involved in the sialic acid binding site [36,51]. To identify the effects of the VP3-I88V mutation, we examined the simulated VP3-I88 and VP3-I88V structures (see Methods). To gather information on the dynamics, longer simulations are required; nonetheless, our simulations provide useful insights into structural changes upon mutating isoleucine 88 to valine in VP3. In Figure 9C, we display the final frame of the simulations of VP3-I88 (red) and VP3-I88V (pink) structures. Compared to the VP3-I88 structure, the mutated structure exhibits important changes in the CD and GH loops. In the CD loop, such changes cause D91 to point away from R104, disrupting this interaction. The GH loop appears to move away from the C and H beta strands, breaking the interaction between K183 and Y191 in the VP3-I88 structure and forming a new one between D87 and N188. Similar conformational changes in the VP3 GH loop occur during particle expansion of the fully native virion and may precede EV uncoating [52]. Finally, the distance between F178 and V88 is 0.5 Å greater than that between F178 and I88, suggesting the major mobility of this residue in the mutated structure. Overall, our structural analysis shows that VP3 I88V mutation affects the network of interactions among protomer residues in an extremely sensitive region of VP3. These structural alterations resemble the conformational changes during EV uncoating, thus suggesting that the I88V mutation could prime the expansion and uncoating of the native virus prematurely and resulting in reduced virulence.

## 4. Discussion

In this study, we found that type I IFNR-ko mice were susceptible to EV-D68 infection and recapitulate the paralysis observed in severe human infection. We then determined that strain 49130 is non-pathogenic (as it causes no observable disease or paralysis) and that the other tested strains are pathogenic in this mouse model. Sequence comparisons revealed several nucleotide and amino acid changes between the pathogenic and non-pathogenic EV-D68 strains. By constructing a series of infectious clones to generate chimeric and point-mutated EV-D68 viruses, an amino acid change at position 88 in VP3 was shown to be the major virulence determinant for the attenuation phenotype, in addition to the other two weaker factors, VP1-L1P and 3A-H47R. Mutation VP3-I88V reduced virus replication in the CNS of mice but showed little effect on that in cell lines, including RD, SK-N-SH and Neuro-2A cells.

We noticed some sequence discrepancies for US/MO/14-18949 between our infectious cDNA clone (49130) and the two records from GenBank, KM851227 and MH708882. Among them, two differences are relevant to our study: amino acid position 88 in VP3 (VP3-88) and position 1 in VP1 (VP1-1). The nucleotide sequence encoding the VP3-V88 is “GTA” in our infectious cDNA clone and KM851227 but “ATA” in MH708882; the codon is “CCA” for VP1-P1 in our infectious cDNA clone and MH708882 but “CTA” in KM851227. Since we found that the amino acid change from isoleucine to valine at VP3-88 (VP3-I88V) and leucine to proline at VP1-1 (VP1-L1P) attenuates EV-D68 virulence in our mouse model, this sequence discrepancy may cause confusion. We constructed infectious cDNA clones using viral RNA extracted from viruses from BEI Resources without further passage in cell culture and our approach did not involve modifications at these two positions, so we believe it is highly unlikely that the two variations were introduced during the cloning. Thus, we continued our study with the infectious clones constructed.

Our finding of VP3-I88V as a major virulence determinant for EV-D68 in mice is consistent with a recent report in which a non-pathogenic EV-D68 strain US/MO/14-18949 (49130 in our study) became pathogenic after 30 passages in AG129 mice [28]. A change from valine to isoleucine at amino acid 88 in VP3 in the mouse-adapted virus was reported as one of the potential mutations contributing to the increased virulence. Here, we demonstrate that, by changing the same amino acid of a pathogenic strain to yield an attenuated mutant virus, another potential virulence determinant VP1-L1P was revealed in their mouse adaptation and our virulence test. However, we found the nucleotides coding VP1-P1 as “CCA” in our infectious clone 49130, which is different from “CTA” that encodes leucine in the GenBank sequence (accession number: KM851227), as discussed above. This VP1-L1P mutation was found in the mouse-adapted 49130 with increased virulence in mice, but the same VP1-L1P in our study partially attenuated the virulence of strain 49131 in mice. We also examined the prevalence of this VP3-I88V in clinical isolates by analyzing EV-D68 sequences from the Virus Pathogen Resource (ViPR) [53] and found that 674 of the 676 collected sequences have isoleucine but only two have valine at position 88 of VP3. As viruses face higher selection pressures when disseminating from tissues to tissues in infected animals, the dominant mutations in the adapted virus population may help to increase viral fitness, adapt quickly to different micro-environments or antagonize immune responses in vivo.

Analysis of the structure through the simulations showed that the mutation of isoleucine 88 to valine in VP3 caused important rearrangements in the CD and GH loops and broke the native network of interactions among VP3 residues in a region that is crucial for viral activities because it has been identified as the sialic acid binding site [36,51]. In the work by Y. Liu et al. [51], the crystal structure of EV-D68, in association with sialylated glycan receptor analogues, shows that the sialic acid moiety of these ligands binds to the virus canyon by making interactions with several residues located in proximity to I88 and causes conformational changes in the loops connecting the sialic acid binding site and the VP1 hydrophobic pocket and the canyon, apparently determining the expulsion of the pocket factor. In a more recent study [36], the authors identified sulfated glycosaminoglycans as receptors in absence of sialylated glycans and, by performing structural analysis, confirmed the binding site of the sialic acid in the canyon and the displacement of the pocket factor. Our simulations show that I88 is located right under the sialic acid binding site, and the structural changes caused by the I88V mutation likely facilitate the accommodation of the sialic acid and reduce the enthalpic cost of its binding. Interestingly, the motion of the VP3 CD loop along with that of the VP3 GH loop and VP1 GH loop upon binding of sialylated receptor analogues provokes displacement of the pocket factor site, the mechanism suggested to be initiated by the electrostatic interactions between Q89 and the N-acetylneuraminic acid [51]. Even if our systems lack the pocket factor, the conformational changes in the simulated VP3-I88V structure with respect to the VP3-I88 structure suggest that the I88V mutation destabilizes the virus in a similar way when sialic acid binds and the pocket factor is ejected to prime the virus for uncoating [52]. In addition, cryo-EM information suggests that the VP3 GH loop motion can be considered one of the molecular determinants involved in the process of EV uncoating [52]. Our structural analysis shows that the I88V mutation affects the native structure of the VP3 GH loop in a manner consistent with our experimental results. However, because the VP3 GH loop is located at one of the protomer corners in contact with VP2 of another protomer and at the interface between two different pentamers, longer simulations with at least two pentamers would be required in order to fully examine the loop dynamics and its effects on the interactions between pentamers and to formulate broad conclusions on the whole EV uncoating process.

Capsid proteins of EV-D68 may have undiscovered functions or interactions with host proteins that affect virus replication, infectivity or virulence [54,55]. Another proposed mechanism of interaction involves the neuro-expressed intracellular adhesion molecule 5 (ICAM-5). ICAM-5 may be required for successful EV-D68 infection [56]. The pre-2014 Fermon strain requires sialic acid, but the strains 49129 and 49132 (US/KY/14-18953, not included in this study) can infect cells in the absence of sialic acid [56]. ICAM-5 neutralizes EV-D68 in vitro by catalyzing the conversion of mature virions to A-particles [57]. While attempts to determine the structure of the EV-D68 -ICAM-5 complex have been unsuccessful, structures of the complex of EV-D68 and neutralizing antibodies that also precipitate the A-particle state bind with the BC, DE, EF and HI loops of VP3 [57]. Both EV-D68 strains 49130 and 49131 grow in Neuro-2A cells with or without neuraminidase treatment, which removes sialic acid from the cell surface, suggesting that differential virulence in mouse models may be due to another cellular factor [58]. If cell factors, such as ICAM-5, are involved in neurovirulence, there may be functional differences in the interaction between murine and human ICAM-5 and their interactions with varying strains of EV-D68 that determine successful infection. This interaction may be blocked by the VP3-I88V and thus prevent virus replication in mice. To validate this hypothesis, expression levels of viral proteins and viral RNA copy number in mouse tissues may be useful indicators. If this hypothesis holds, identification of this host factor may be important as it could serve as a target for antiviral development.

The combination of reverse genetics and a small animal model provides a straightforward approach for the ferreting of virulence determinants of pathogens. As the Severe Acute Respiratory Syndrome Coronavirus 2 (SARS-CoV-2) is devastating the globe, knowledge obtained from such studies can be valuable. Rapid reconstruction of SARS-CoV-2 [59] and susceptible receptor (human angiotensin-converting enzyme 2, hACE2)-transgenic mouse models have been reported [60,61], making such studies possible. However, difficulties are expected. The SARS-CoV-2 strain HB-01 causes only slight bristled fur and body weight loss in hACE2 transgenic mice but not other clinical symptoms [60]. If a SARS-CoV-2 strain that causes virulence greater than the HB-01 strain is not available, an alternative can be to generate mouse-adapted viruses that display increased virulence [62,63,64]. In addition, bioinformatic analysis has been applied to elucidate potential viral determinants for pathogenicity. Similar to virulence, Gussow et al. analyzed coronaviruses of high and low case fatality rate (CFR), and reported features of enhanced nuclear localization signal (NLS) in the nucleocapsids and insertions in the spike protein as shared by the high CFR coronaviruses [65]. Results from such bioinformatic analysis could provide valuable candidates to be further validated in animal models. Even with these limitations, knowledge obtained from such studies could still help to further our understanding of viral pathogenicity and thus promote the development of antivirals and vaccines.

## 5. Conclusions

In conclusion, we report an amino acid change from isoleucine to valine at position 88 in VP3 of EV-D68 as a major virulence determinant in a type I IFNR-deficient mouse model. This VP3-I88V mutation reduces EV-D68 virulence and virus replication in the brain and spinal cord of mice but has little effect on virus replication in cultured cells. The VP3-I88V completely attenuated the pathogenic strain 49131 by improving the survival rate for infected mice from 0% to 100% and also removed its neurovirulence, based on results from tissue distribution analysis. However, virus replication in cell lines, including human muscular and neuronal cells and mouse neuronal cells, was not affected by the VP3-I88V. Our structural analysis shows that the I88V mutation generates conformational changes in the native structure that are consistent with structural intermediates observed experimentally during the EV uncoating process. However, the mechanism for VP3-I88V attenuation in mice remains unknown. One clear difference between cell culture and infection in animals is the immune system. While we used animals that were defective in type I INF responses, VP3-I88 could also suppress other arms of the antiviral defense mechanism. Further studies are required in order to determine the mechanism of action that determines neurovirulence. In addition, infectious cDNA clones for EV-D68 constructed in this study can be great tools for evaluating the effect of individual mutation on virus phenotypes.

## Figures and Tables

**Figure 1 viruses-12-00867-f001:**
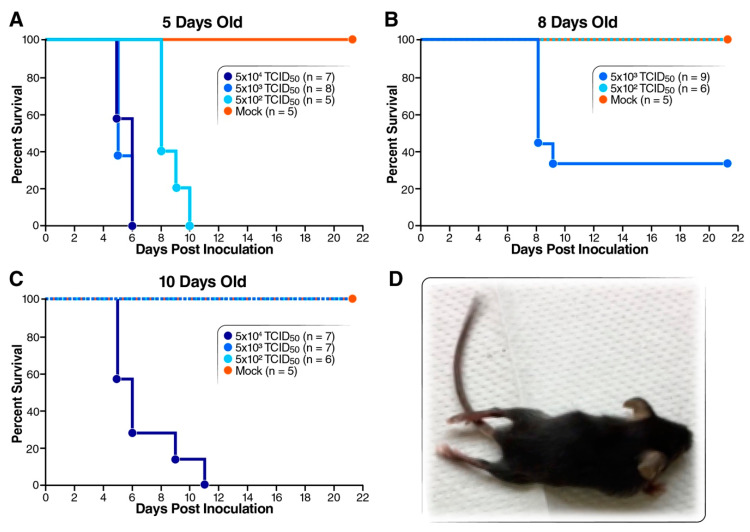
Susceptibility of type I interferon receptor deficient mice to EV-D68 infection. EV-D68 49129 used in this experiment was from cell culture. (**A**–**C**) Mice of 5, 8 and 10 days old were i.p. inoculated with 5 × 10^2^–5 × 10^4^ TCID_50_ of EV-D68 strain 49129 (US/MO/14-18947) and monitored for survival for 21 days. Mice in the control group were injected with 100 µL of viral medium. (**D**) Paralysis on both posterior limbs was observed after i.p. inoculation of strain 49129.

**Figure 2 viruses-12-00867-f002:**
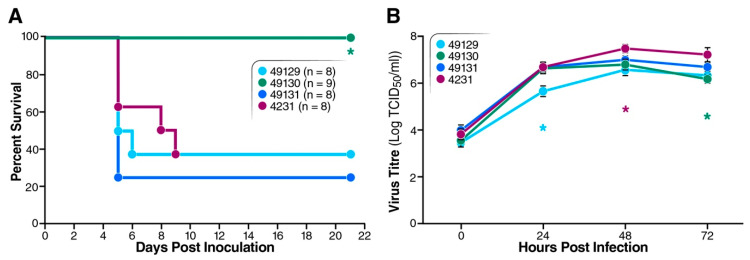
Characteristics of various EV-D68 strains. (**A**) Virulence test for various EV-D68 strains. Seven-day-old mice were i.p. inoculated with 100 uL of virus delivering 1 × 10^4^ TCID_50_ of EV-D68 strain NR-49129, NR-49130, NR-49131 or CA/14-4231. Survival of the mice was monitored for 21 days. Asterisk (*) represents significantly different survival curve, comparing to that of 49131-infected mice (log-rank test, *p* < 0.05). (**B**) Replication in RD cells. RD cells were infected with EV-D68 strains tested in Figure 2A at m.o.i. of 0.01 and harvested at 0, 24, 48 and 72 h for TCID_50_ to determine virus titers. Results of triplicates are shown as mean ± S.D. Asterisks represent significant differences in virus titers compared with 49131 (Student’s *t* test, *p* < 0.05).

**Figure 3 viruses-12-00867-f003:**
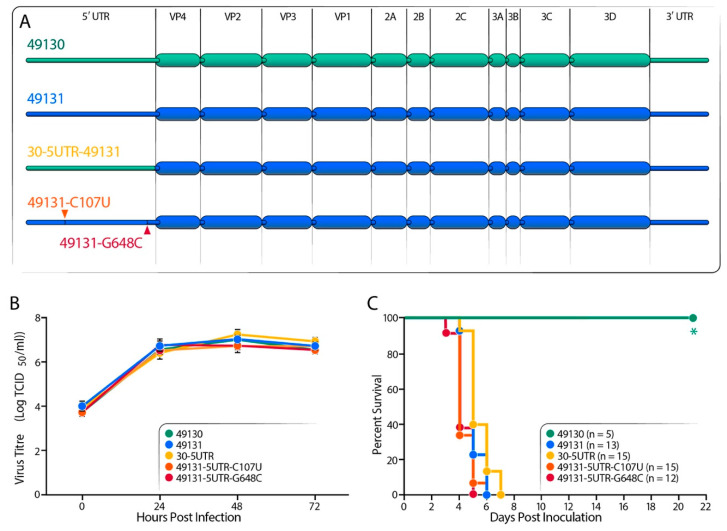
Virulence test for 5′-UTR-modified EV-D68 viruses. (**A**) Schematic presentation of the gene-swapped and point-mutated viruses. Strains 49131 and 49130 were used as backbone to generate these 5′-UTR modified viruses. Strain 30-5UTR-49131 is 49131 carrying 5′-UTR from 49130. Nucleotide changes U107C and G648C were individually introduced into 49131. (**B**) Replication analysis for the engineered EV-D68 viruses was performed with RD cells, as described in Figure 1B. Results of triplicates are shown as mean ± S.D. No significant difference in virus replication ability compared with strain 49131 was detected by Student’s *t* test (*p* < 0.05). (**C**) Seven-day-old Tg21/IFNR-ko mice were i.p. inoculated with 1 × 10^5^ TCID_50_ of virus and monitored for 21 days to compare the virulence of the 5′-UTR-modified EV-D68 viruses. Viruses 49131, 30-5UTR (30-5UTR-49131), 49131-5UTR-U107C and 49131-5UTR-G648C were tested with 2 litters of pups. Asterisk represents significantly different survival curve comparing with that of 49131-infected mice detected by using log-rank test (*p* < 0.05).

**Figure 4 viruses-12-00867-f004:**
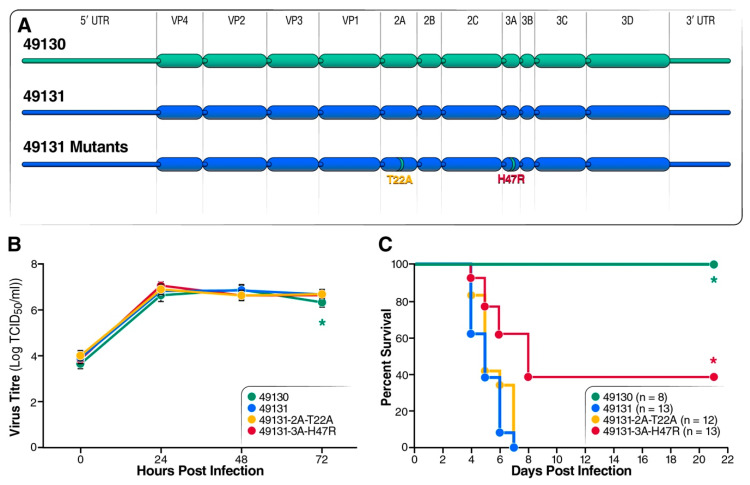
Validation of genetic changes in the non-structural genes on EV-D68 virulence with strain 49131 as a backbone. (**A**) Schematic of the gene-swapped and point-mutated viruses. Strain 49131 was used as backbone to generate these recombinant viruses. Amino acid changes 2A-T22A and 3C-H47R were introduced into 49131 individually. (**B**) Replication analysis for the engineered EV-D68 viruses carrying mutations in the non-structural region (NSR) was performed with RD cells, as described in Figure 1B. Results of triplicates are shown as mean ± S.D. Asterisk: significantly lower virus titer of 49130 at 72 h post infection comparing with 49131 was detected by Student’s *t* test (*p* < 0.05). (**C**) Seven-day-old mice were i.p. inoculated with 1 × 10^5^ TCID_50_ of viruses carrying 2A-T22A or 3C-H47R and monitored for 21 days. 49131, 2A-T22A, and 3C-H47R were tested in 2 litters of pups. Asterisks represent survival curves which, compared with those of 49131-infected mice, were determined to be significantly different by using log-rank test (*p* < 0.05).

**Figure 5 viruses-12-00867-f005:**
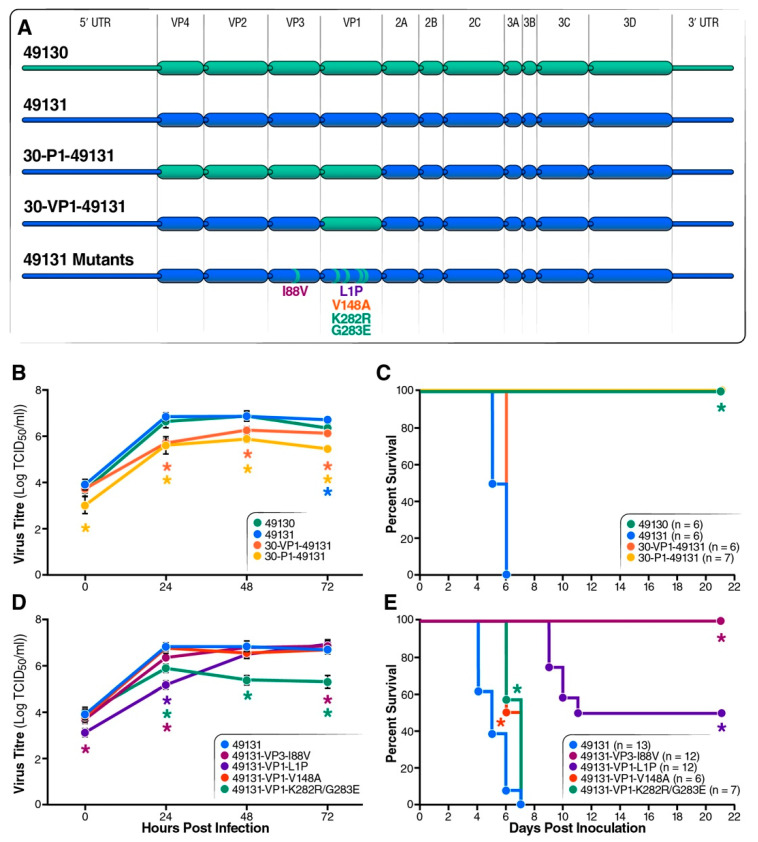
Validation of the conserved amino acid changes in structural proteins on replication and virulence of EV-D68. Asterisks represent significant difference as compared with 49131. (**A**) Representation of the gene composition for viruses carrying swapped VP1 or P1 gene and amino acid substitutions in the structural gene. Strain 49131 was used as backbone to generate these recombinant viruses. Amino acid changes VP3-I88V, VP1-L1P and VP1-V148A were introduced into 49131 individually and VP1-K282R and VP1-G283E were changed together. (**B**) Replication analysis for VP1 or P1 swapped viruses, as shown in Figure 1B. (**C**) Seven-day-old mice were inoculated i.p. with 1 × 10^5^ TCID_50_ of strain 49131 carrying VP1 or P1 gene from 49130 and monitored for 21 days. (**D**) Replication analysis for viruses carrying mutations in the structural genes, as shown in Figure 1B. (**E**) Virulence test for viruses carrying mutations in the structural genes were performed by i.p. injecting 7-day-old mice with 1 × 10^5^ TCID_50_ of virus and monitored for 21 days. In this experiment, 49131, VP3-I88V and VP1-L1P were tested with 2 litters of pups. (**B**,**D**) Results of triplicates are shown as mean ± S.D. Asterisks represent significantly different virus titers of mutant viruses compared with 49131 detected by Student’s *t* test (*p* < 0.05). (**C**,**E**) Asterisks represent significantly different survival curves, compared with that of 49131-infected mice, as detected by log-rank test (*p* < 0.05).

**Figure 6 viruses-12-00867-f006:**
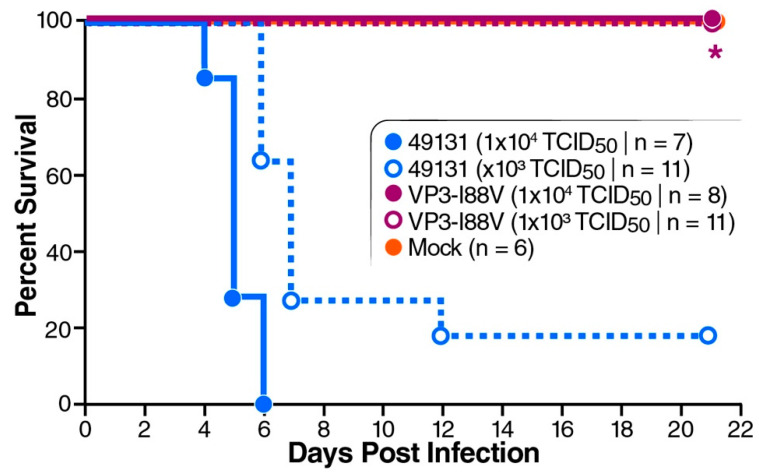
Neurovirulence test for 49131-VP3-I88V mutant. Five-day-old mice were i.c. inoculated with 10 µL of inoculum delivering 1 × 10^3^ or 1 × 10^4^ TCID_50_ of 49131 or 49131-VP3-I88V virus. Mice in control group were injected with 10 µL of viral medium. Survival was monitored for 21 days. Asterisk represents significantly different survival curves of VP3-I88V-infected mice, compared with that of 49131 (lower dose)-infected mice, as detected by log-rank test (*p* < 0.05).

**Figure 7 viruses-12-00867-f007:**
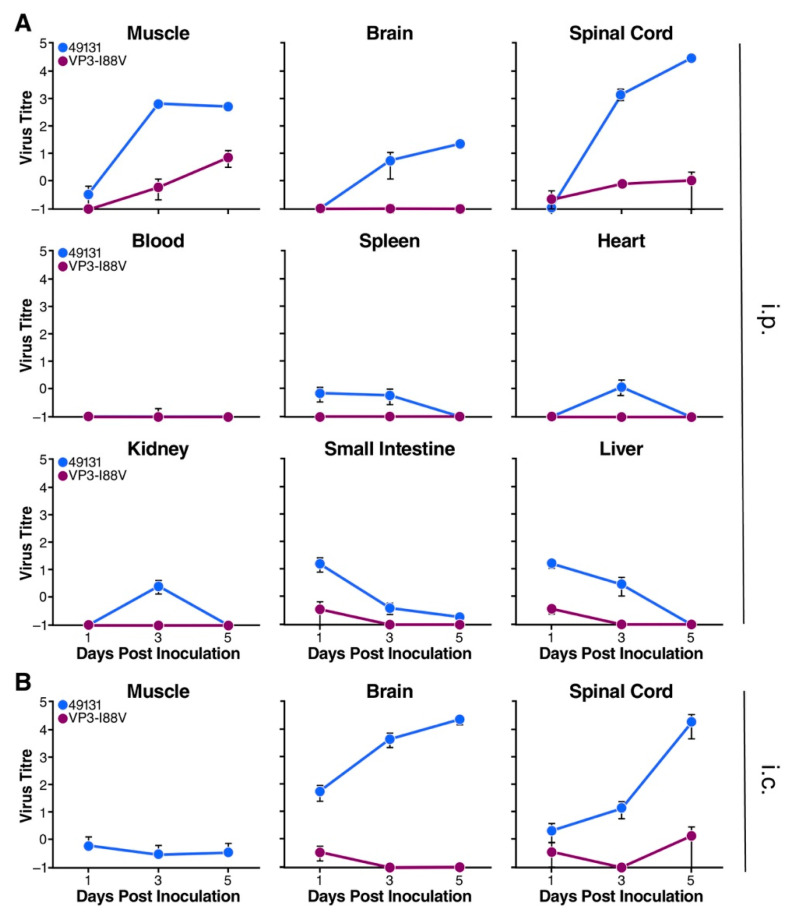
Tissue distribution of EV-D68 virus in Tg21/IFNR-ko mice. Virus titers were determined by a standard TCID_50_ assay and shown as log TCID_50_ per milligram of tissue. Results are shown as mean ± S.E.M. (**A**) Mice of seven days old were i.p. inoculated with 1 × 10^5^ TCID_50_ of 49131 (shown as black) or 49131-VP3-I88V (shown as blue). At days 1, 3 and 5, tissues of three mice from each group were harvested to determine virus titers. (**B**) Mice of 5 days old were i.c. inoculated with 1 × 10^4^ TCID_50_ of 49131 or 49131-VP3-I88V. At days 1, 3 and 5, tissues of three mice from each group were harvested to determine virus titers.

**Figure 8 viruses-12-00867-f008:**
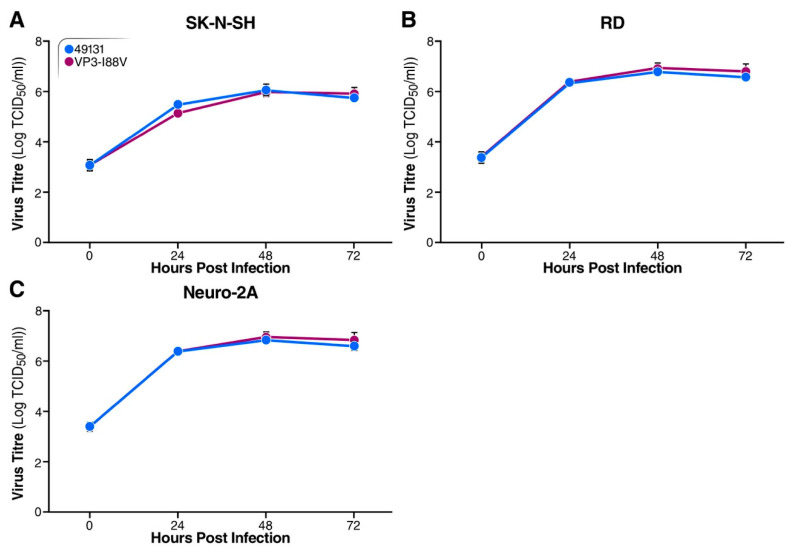
Replication analysis for 49131 and 49131-VP3-I88V in cell lines. SK-N-SH (**A**), RD (**B**) and Neuro-2A (**C**) cells were infected with 49131 or VP3-I88V at an m.o.i. of 0.01 in triplicate and harvested at indicated time points for TCID_50_ assay to determine virus titers. Results of triplicates are shown as mean ± S.D.

**Figure 9 viruses-12-00867-f009:**
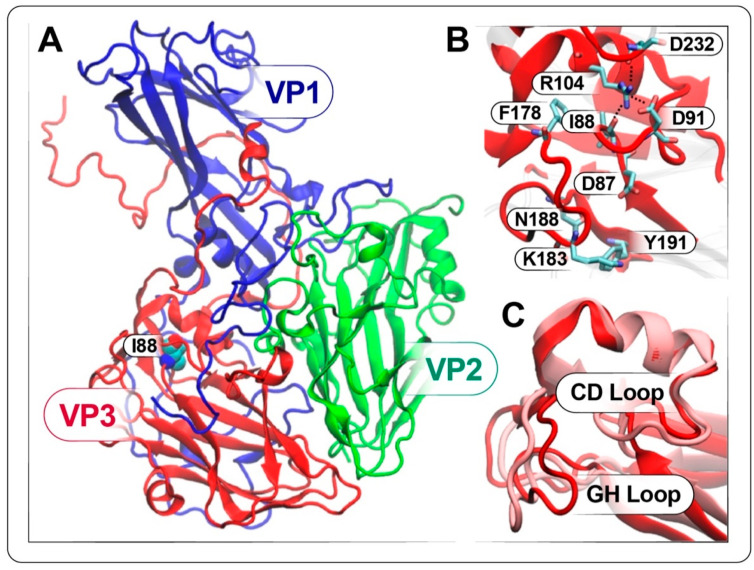
Comparison of the simulated VP3-I88 and VP3-I88V structures. (**A**) VP1 (blue), VP2 (green) and VP3 (red) of EV-D68 are represented in new cartoon format. I88 is represented in van der Waals sphere format. The inset reports on the position of I88 within the pentamer with respect to VP1, VP2 and VP3. (**B**) Representation of the VP3 residues within 12 Å of I88 in new cartoon format. Residues interacting with I88 or showing relevant changes between VP3-I88 and VP3-I88V simulated structures are displayed in licorice format. (**C**) Comparison between the VP3-I88 (red) and VP3-I88V (pink) simulated structures extracted from the last frame of the simulations. In all panels, residues are colored according to the atoms: oxygen in red, carbon in cyan, nitrogen in blue; hydrogens are omitted for clarity.

**Table 1 viruses-12-00867-t001:** EV-D68 strains used in this study.

Strain Name	BEI Resources Catalog Number	Virulence in IFNR-Ko Mice
US/MO/14-18947	NR-49129	Death
US/MO-14-18949	NR-49130	No observable disease
US/IL-14-18952	NR-49131	Death
CA/14-4231	Not Applied	Death

**Table 2 viruses-12-00867-t002:** Conserved nucleotide and amino acid differences in virulent and non-virulent EV-D68 strains.

Region	Position ^1^	Sequence in Virulent Strains ^2^	Sequence in Non-Virulent Strain ^2^
5′-UTR	107	Cytosine (C)	Uridine (U)
	648	Adenine or Guanine (A, G) ^3^	Cytosine (C)
VP3	88	Isoleucine (I)	Valine (V)
VP1	1	Leucine, Isoleucine (L, I) ^4^	Proline (P)
	148	Valine (V)	Alanine (A)
	282	Lysine (K)	Argine (R)
	283	Glutamine, Glycine or Lysine (Q, G, K) ^5^	Glutamic Acid (E)
2A	22	Threonine (T)	Alanine (A)
3A	47	Histidine (H)	Arginine (R)

^1^: Numbers refer to each region or gene. Nucleotide for 5′-UTR; amino acid for viral proteins. ^2^: Virulence as shown in Table 1. ^3^: Adenine in strains NR-49129 and CA/14-4231; guanine in strain NR-49131. ^4^: Leucine in strains NR-49129 and NR-49131, isoleucine in strain CA/14-4231. ^5^: Glutamine in strain NR-49129, glycine in strain NR-49131 and lysine in strain CA/14-4231.

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
