# Peer review of "Mapping Attenuation Determinants in Enterovirus-D68"

_viruses, 2020, doi:10.3390/v12080867_

Round 1
Reviewer 1 Report
Manuscript ID viruses-878922
Yeh et al describe the mapping of determinants in Enterovirus-D68 (EV-D68) implicated in the attenuation of the virus neurovirulence in neonatal Type I interferon receptor knockout mice.
Comments:
The manuscript is nicely written, easy to understand, and the methodology looks fine.
In the material and method, the authors should mention which method has been used to calculate the TCID50 titers (Karber method… ?)
In the results, figure 1B, why the 50,000 TCID50 inoculation dose has not been tested for the 8 days old mice, at the contrary of the 5 and 10 days old. Similarly, did the susceptibility of 5 to 10 days old mice to the other strains (49130, 49131, 4231) has been tested at 500 to 50,000 TCID50?
In Figure 2A, there is a difference between the text (line 211) referring to 8 days old mice and the legend of the figure referring to 7 days old mice. Why 7 days old mice have been used, instead of 8 days old mice, like in Figure 1?
In figure 2B, 3B, 4B, 5B, 5D, and 8, the number of independent replicates used for the mean/SD calculation is missing.
In Figure 6, there is a difference with the text (line 332-333). According to the figure, we have for EV-D68/49131 0% survival rate with 1,000 TCID50 and 18.2% survival rate for 10,000 TCID50. According to the text (and more logically), it is the contrary.
The authors use in the discussion the term “(non)pathogenic” to make the distinction between the EV-D68 49130 and the others. However, the only pathogenic phenotype studied is the lethality/paralysis of the virus strains. Neuropathogeny or lethality would be more appropriate.
Author Response
We would like to thank the editor and reviewers of Viruses for their time and suggestions, which has helped us improve the quality of the present revision. The current revised manuscript attempts to incorporate all the valuable points.
Reviewer #1:
Specific comments:
- In the material and method, the authors should mention which method has been used to calculate the TCID50 titers (Karber method… ?)
Ans.: We used the Spearman & Karber method as described in Virology Methods Manual (Hierholzer & Killington, 1996) to calculate TCID50. We have now added this information to the Materials and Methods section and the citation in the current revision (Line 180).
- In the results, figure 1B, why the 50,000 TCID50 inoculation dose has not been tested for the 8 days old mice, at the contrary of the 5 and 10 days old. Similarly, did the susceptibility of 5 to 10 days old mice to the other strains (49130, 49131, 4231) has been tested at 500 to 50,000 TCID50?
Ans.: We performed this experiment to test whether this mouse strain is susceptible to EV-D68 infection. Although we didn’t get enough 8-day-old mice to include the dose of 5x104 TCID50 of virus at that time, we confirmed that this mouse model as susceptible to EV-D68 infection and thus proceeded with the mapping of virulence determinants.
We did not test susceptibility of 5- and 10-day-old mice to the other EV-D68 strains with the same doses. We only tested virulence of those EV-D68 strains using 7-day-old mice as shown in Figure 2. We found one strain (49130) that did not cause observable disease (signs of ruffled hair, hunched back, reduced mobility and paralysis) while the others caused paralysis and death, and thus focused on mapping virulence factors for the phenotype. However, we think it would be interesting to compare the median lethal dose (LD50) of the strains 49129, 49131, and 4231 as they may display distinct virulence at other doses and reveal other minor or strain-specific determinants.
- In Figure 2A, there is a difference between the text (line 214) referring to 8 days old mice and the legend of the figure referring to 7 days old mice. Why 7 days old mice have been used, instead of 8 days old mice, like in Figure 1?
Ans.: We used 7-day-old mice in this experiment and corrected this discrepancy in the revised manuscript (Line 288).
As Reviewer indicated, we didn’t test the other strains as described in Figure 1. We decided to use 7-day-old based on our analysis to Figure 1. Result in Figure 1 suggested that the susceptibility (death and time of death onset) of Tg21/IFNR-ko mice to EV-D68 infection is dependent on the age of mice and the injected viral dose. We found the 5-day-old mice were too young to reveal the distinct virulence as all mice injected with 5x103 and 5x104 TCID50 of virus died between Days 5-6 post infection, and 10-day-old mice were susceptible only to the highest tested dose (5x104). For 8-day-old mice, the earliest death onset was observed at Day 8 post infection, so we believe 7-day-old mice infected with higher virus dose should be the optimal condition to observe death onset and survival of infected mice.
- In figure 2B, 3B, 4B, 5B, 5D, and 8, the number of independent replicates used for the mean/SD calculation is missing.
Ans.: We added this information (results of triplicates) in the revised manuscript.
- In Figure 6, there is a difference with the text (line 332-333). According to the figure, we have for EV-D68/49131 0% survival rate with 1,000 TCID50 and 18.2% survival rate for 10,000 TCID50. According to the text (and more logically), it is the contrary.
Ans.: Yes, the lines and symbols for 1x104 and 1x103 of 49131 in Figure 6 were reversed. We corrected this discrepancy in the revised manuscript.
- The authors use in the discussion the term “(non)pathogenic” to make the distinction between the EV-D68 49130 and the others. However, the only pathogenic phenotype studied is the lethality/paralysis of the virus strains. Neuropathogeny or lethality would be more appropriate.
Ans.: We added ‘(as it causes no observable disease or paralysis)‘ to this sentence to define ‘nonpathogenic’ (Line 539). We also described the signs of disease onset that were monitored to Section 2.6 in the Material and Methods as below:
‘Infected mice were monitored daily for signs of disease onset including ruffled hair, hunched back, reduced mobility and paralysis.’ (Line 190)
Reviewer 2 Report
This is an exemplary study of pathogenetic events in the life of an emerging enterovirus. Molecuar and in vivo experiments are well interwoven, as it should be for advancing scientific knowledge. The study is enriched by structural investigations. Tables, Figures, References are OK. Very minor spelling errors must be checked out.
A few notes:
1) The genetic background of the employed mouse strains should be given in full
2) the tissue origin (and characterizing markers when available) should be inserted into the Mat & Meth section
3) Viral titers should be reported as 5x102, 5x104 etc. instead than arithmetic values - This both in the text, Figures and Tables
Line 167: charm-gui (please define more completely)
Line 186: every 20 ps (please define as picoseconds?)
Line 220: efficiently in RD cells: unsatisfactory. Give a comparison of actual titers
Line 468: sialic acid binding site... Please make it clear (summarize) the main conclusions of the two studies
Line 516: I could not understand well the sentence: VP3-188V may not suppress... Should it be: could also suppress... Please make your idea clear.
A final note:
I would appreciate if a consideration could be inserted regarding the current difficulty of interpreting SARS-CoV2 factors and mutations in pathogenesis. This is a new virus, experience teaches that even detecting pathogenicity determinants in long-known viruses is extremely difficult.
Author Response
Reviewer #2:
Specific comments:
- The genetic background of the employed mouse strains should be given in full
Ans.: The Tg21/IFNR-ko mouse strain was derived from C57BL/6. We added this information to the revised manuscript (line 141).
- the tissue origin (and characterizing markers when available) should be inserted into the Mat & Meth section
Ans.: We added this information to the Material and Methods section (Line 199). We did not characterize these tissues with characterizing markers.
3) Viral titers should be reported as 5x102, 5x104 etc. instead than arithmetic values - This both in the text, Figures and Tables
Ans.: We changed the format in the figures and text as suggested in the revised manuscript.
Line 167: charm-gui (please define more completely)
Ans.: We added the following text to address this question.
(Lines 229-236) A number of residues from the crystal structure [36] in the beta sheets connecting link were unresolved and we modeled these into the structure using CHARMM-GUI software [37]. We provided as input to CHARMM-GUI software the pdb-format structure file downloaded from the Protein Data Bank website (https://www.rcsb.org/), we patched the N-terminus of each capsid protein protomer with an acetylated (ACE) group and the C-terminus with a standard C-terminus patching group, and modeled the missing residues by checking a box in CHARMM-GUI (CHARMM-GUI uses GalxyFill). To generate the mutated structure, we repeated this protocol and selected the "Mutation" box within CHARMM-GUI.
Line 186: every 20 ps (please define as picoseconds?)
Ans.: We modified this in the current revision (line 256). We also corrected the ‘ns’ (line 243) and the ‘fs’ (line 253) in this section.
Line 220: efficiently in RD cells: unsatisfactory. Give a comparison of actual titers
Ans.: We corrected this in the current revision. (Lines 297-298, 338, 386-387, 399-404)
Line 468: sialic acid binding site... Please make it clear (summarize) the main conclusions of the two studies
Ans.: We added the following text to this section to address this question.
Lines 581-591. Analysis of the structure through the simulations showed that the mutation of isoleucine 88 to valine in VP3 caused important rearrangements in the CD and GH loops and broke the native network of interactions among VP3 residues in a region crucial for viral activities because it has been identified as the sialic acid binding site [36, 51]. In the work by Y. Liu et al. [51], the crystal structure of EV-D68 in association with sialylated glycan receptor analogues shows that the sialic acid moiety of these ligands binds to the virus canyon by making interactions with several residues located in proximity of I88 and causes conformational changes in the loops connecting the sialic acid binding site and the VP1 hydrophobic pocket and the canyon, apparently determining the expulsion of the pocket factor. In a more recent study [36], the authors identified sulfated glycosaminoglycans as receptors in absence of sialylated glycans and, by performing structural analysis, confirmed the binding site of the sialic acid in the canyon and the displacement of the pocket factor. Our simulations show that I88 is located right under sialic acid binding site, and the structural changes caused by the I88V mutation likely facilitate accommodation of the sialic acid and reduce the enthalpic cost of its binding.
Line 516: I could not understand well the sentence: VP3-188V may not suppress... Should it be: could also suppress... Please make your idea clear.
Ans.: The EV-D68 strain 49131 that has the isoleucine at the 88th amino acid position of VP3 causes paralysis and death in our mouse model while the 49131-VP3-I88V that has an amino acid substitution from isoleucine to valine at this position is totally attenuated. The mouse model strain used in this study is defective in type I interferon response, but we can’t exclude the possibility of VP3-I88 targeting other components of the immune system to cause paralysis and death. We modified the sentence to ‘While we used animals defective in type I INF responses, VP3-I88 could also suppress other arms of the antiviral defense mechanism.’ (lines 657-658) to make it clearer.
A final note:
I would appreciate if a consideration could be inserted regarding the current difficulty of interpreting SARS-CoV2 factors and mutations in pathogenesis. This is a new virus, experience teaches that even detecting pathogenicity determinants in long-known viruses is extremely difficult.
Ans.: We added a section to Discussion as below.
Lines 627-644. ‘The combination of reverse genetics and small animal model provides a straightforward approach for the ferreting of virulence determinants of pathogens. As the Severe Acute Respiratory Syndrome Coronavirus 2 (SARS-CoV-2) is devastating the globe, knowledge obtained from such studies can be valuable. Rapid reconstruction of SARS-CoV-2 [59], and susceptible receptor (human angiotensin-converting enzyme 2, hACE2)-transgenic mouse models have been reported [60, 61], making such studies possible. However, several difficulties and limitations are expected. The SARS-CoV-2 strain HB-01 causes only slight bristled fur and body weight loss in the hACE2 transgenic mice, but not other clinical symptoms [60]. If a SARS-CoV-2 strain that causes virulence greater than the HB-01 strain is not available, a common solution can be to generate mouse-adapted viruses that display increased virulence [62-64]. In addition, bioinformatic analysis has been applied to elucidate potential viral determinants for pathogenicity. Similar to virulence, Gussow et al. analyzed coronaviruses with high and low case high fatality rate (CFR), and reported features of enhanced nuclear localization signal (NLS) in the nucleocapsids and insertions in the spike protein as shared by the high CFR coronaviruses [65]. Results from such bioinformatic analysis could provide valuable candidates to be further validated in animal models. Even with these limitations, knowledge obtained from such studies could still help further our understanding of viral pathogenicity and thus are important in the race of developing antivirals and vaccines.’
Other modifications by the authors:
- Lines 49, 56, 7665, and 666. The name of one author, ‘Benjamin Adam Catching’ was changed to ‘Adam Catching’ per author’s request for consistency.
- Lines 131-132. We added ‘Infectious cDNA clones of EV-D68 strains 49129, 49130, and 49131 have been banked in BEI Resources (catalog numbers: NR-52009, NR-52010 and NR-52011).’ to section 2.1 in the Materials and Methods.
- Line 191. We added signs of disease onset that were monitored to section 2.6 in the Materials and Methods.
- Figure 2A, Figure 4B and Figure 5E. Asterisks were missed, and corrected in the revision.
- Figure 5D and 5E. Mutant names were modified to be consistent with the text.